# Development and Characterization of a Factor V-Deficient CRISPR Cell Model for the Correction of Mutations

**DOI:** 10.3390/ijms23105802

**Published:** 2022-05-22

**Authors:** Luis Javier Serrano, Mariano Garcia-Arranz, Juan A. De Pablo-Moreno, José Carlos Segovia, Rocío Olivera-Salazar, Damián Garcia-Olmo, Antonio Liras

**Affiliations:** 1Department of Genetics, Physiology and Microbiology, School of Biology, Complutense University, 28040 Madrid, Spain; luisserr@ucm.es (L.J.S.); jdepablo@ucm.es (J.A.D.P.-M.); 2Fundación Jiménez Díaz University Hospital Health Research Institute, Fundación Jiménez Díaz University Hospital, 28040 Madrid, Spain; mariano.garcia@quironsalud.es (M.G.-A.); rocio.olivera@quironsalud.es (R.O.-S.); damian.garcia@uam.es (D.G.-O.); 3Department of Surgery, Autonomous University of Madrid, 28049 Madrid, Spain; 4Differentiation and Cytometry Unit, Innovative Hematopoietic Therapies Division, Center for Energy, Environmental and Technological Research (CIEMAT), Biomedical Research Networking Center for Rare Diseases (CIBERER), 28040 Madrid, Spain; jc.segovia@ciemat.es

**Keywords:** gene therapy, gene editing, CRISPR, factor V deficiency, coagulopathies, rare diseases

## Abstract

Factor V deficiency, an ultra-rare congenital coagulopathy, is characterized by bleeding episodes that may be more or less intense as a function of the levels of coagulation factor activity present in plasma. Fresh-frozen plasma, often used to treat patients with factor V deficiency, is a scarcely effective palliative therapy with no specificity to the disease. CRISPR/Cas9-mediated gene editing, following precise deletion by non-homologous end-joining, has proven to be highly effective for modeling on a HepG2 cell line a mutation similar to the one detected in the factor V-deficient patient analyzed in this study, thus simulating the pathological phenotype. Additional CRISPR/Cas9-driven non-homologous end-joining precision deletion steps allowed correction of 41% of the factor V gene mutated cells, giving rise to a newly developed functional protein. Taking into account the plasma concentrations corresponding to the different levels of severity of factor V deficiency, it may be argued that the correction achieved in this study could, in ideal conditions, be sufficient to turn a severe phenotype into a mild or asymptomatic one.

## 1. Introduction

Factor V (FV), also called labile factor or pro-accelerin, is a coagulation factor that plays a key role in the coagulation cascade. In humans, the FV gene (*F5*) has an approximate size of 80 kb, is located in chromosome 1q24.2 and consists of 25 exons and 24 introns. FV cDNA is 6914 bp in length. Eighty percent of circulating FV is produced in the liver by hepatocytes, while the remaining 20% resides in platelet granules [1,2,3,4]. This circulating factor is a glycosylated 330 kDa polypeptide whose activation results in a loss of its B-domain and in the splicing of the protein into a heavy 105 kDa chain, which contains the A1 and A2 domains, and a light 71–74 kDa chain, which contains the A3, C1 and C2 domains. The heavy chain interacts with activated factor X and prothrombin, while the light chain interacts with membrane phospholipids [5,6]. FV deficiency, known as parahemophilia or Owren’s disease, is an autosomal recessive disorder associated with mutations in the *F5* gene and an ultra-rare condition with an incidence of 1–9/1,000,000 live births [7,8,9].

Clinically, FV deficiency is characterized by the occurrence of mild-to-severe bleeding episodes. Different phenotypes of the disease may develop depending on the levels present in the plasma of FV. A severely deficient phenotype is characteristic of plasma levels < 1% of normal, a moderately deficient phenotype is associated with plasma levels between 1% and 10% of normal and a mild/asymptomatic phenotype occurs with plasma levels above 10% of normal [4,10,11]. Bleeding episodes usually start before the age of six and are associated with a heterogeneous spectrum of hemorrhagic manifestations, ranging from mucosal or soft tissue bleeding (i.e., epistaxis or hemarthrosis) to potentially lethal hemorrhages. Abundant nasal and menstrual bleeding are also distinctive features of this deficiency. Profuse bleeding is likewise common during minor and major surgeries, as well as during dental procedures. Hemorrhagic arthropathy, hematomas and cranial and gastrointestinal bleeding are less frequent [4,7,12].

No plasma-derived or recombinant FV concentrates are currently available to treat FV deficiency. The available treatments consist of administering fresh-frozen (virally inactivated) plasma or the use of Octaplas^®^, an alternative solvent/detergent-treated pharmaceutical product. The latter is associated with a high safety profile against emerging pathogens and prions and contains an optimally controlled combination of different coagulation factors, including FV [13,14,15].

Gene editing by CRISPR/Cas9 has been proposed as a potential therapy for several genetic diseases, including FV deficiency. Indeed, given its ability to eliminate mutated exons and normalize the FV protein, CRISPR/Cas9-based gene editing can restore the protein’s original function [16,17]. This strategy, which uses two CRISPR guides (gRNA) to perform two double-strand breaks (DSB), seems to facilitate a so-called non-homologous end joining (NHEJ) repair [18,19,20]. Several authors have studied the efficacy of precise deletion by NHEJ (NHEJ-PD) and analyzed how this process works when the guides are separated by 23–148 bp, in which case, a precise deletion is the most likely event. The use of two guides to delete a defined DNA fragment and alter the frameshift efficiency of the affected locus in a pseudo-controlled way could allow the generation of knockout models for the study of the biology of the cell, thus contributing to correcting pathologic mutations or generating cellular or animal models of rare diseases [21,22]. In this study, a FV-deficient cell model was generated via CRISPR/Cas9 gene editing with a similar mutation to the one exhibited by the studied patient (c.3279G>A, p.Trp1093*).

## 2. Results

As far as the patient is concerned, a familiar segregation analysis indicated that her mother was heterozygous for the already described c.2218C>T, p.Arg740* mutation [23] with FV plasma levels at 63%, indicative of a mild phenotype, with no noteworthy clinical symptoms. Her father was heterozygous for the c.3279G>A, p.Trp1093* mutation [24] and presented with FV plasma levels of 21%, also associated with a mild phenotype without significant symptoms.

To facilitate the availability of FV deficient cells, an attempt was made to mimic one of the mutations described in this patient in a reference hepatic cell line.

### 2.1. NHEJ-PD Generates an FV KO Cellular Model for the HepG2 Cell Line

The non-homologous end-joining precision deletion (NHEJ-PD) strategy was used to generate a KO FV cellular model in a HepG2 cell line (KO HepG2) (Figure 1A,B) and reproduce a mutation similar to c.3279G>A, p.Trp1093*, which is the mutation exhibited by the studied patient (Figure 1C). A specific pair of guides were designed to produce a deletion of 35 bp (between nucleotides 3263 and 3297), modifying the frameshift of the gene sequence and giving rise to a premature stop codon at a short distance from the patient’s c.3263–3298del, p.Met1089* mutation (4 amino acids away). The efficiency of this gene editing procedure was about 60% (59.5%) (Figure 2A), with other minor genetic modifications also being obtained that were not relevant to our study.

Cells that contained this precise sequence were selected through colony-forming units (CFUs), and the DNA of these clones was analyzed by Sanger sequencing. Some clones were shown to exhibit the sought-after mutation (Figure 2B). For the next assays, three c.3263–3298del, p.Met1089* mutation-positive clone cell lines were used, called LO1, LO8 and LO9.

### 2.2. Generation of an FV TT Cellular Model Derived from the KO HepG2 Cell Line

Using the same procedure but with a different pair of guides, a recovered FV cell line (TT HepG2) was generated from KO HepG2 clones LO1, LO8 and LO9 through NHEJ-PD. In this case, a deletion of 49 bp (c.3242–3291del) removed the premature stop codon; the frameshift of the *F5* was restored in the three clones (Figure 1D). Efficiency was around 41.4% ± 1.3 and Sanger sequencing yielded the expected sequences (Figure 2C).

### 2.3. Comparison and Characterization of the Three Cell Lines Obtained: WT HepG2, KO HepG2 and TT HepG2

The three cell lines obtained were compared and characterized. As shown in Figure 1E, the sizes of the PCR products varied in accordance with the sequential deletions carried out. The PCR product was 403 bp for the WT HepG2 line, 368 bp for the three clones of the KO HepG2 cell line and 368 bp and 319 bp for the TT HepG2 cell line, where two different bands were observed, one corresponding to non-edited cells (368 bp) and another to the edited cells that corrected the mutation (319 bp).

As shown in Figure 3A, no differences were observed between the cell lines in terms of size or form. As shown in Figure 3B, the proliferation assay did not reveal any significant differences between the three cell lines. Nor were there any differences observed in the expression of the membrane molecular markers analyzed between the WT HepG2, KO HepG2 and TT HepG2 cell lines (Figure 3C). Immuno-staining showed that all three HepG2 cell lines (WT, KO and TT) produced fluorescence-detectable FV proteins on immunostaining (Figure 4). The hASCs used as negative control did not produce any fluorescent signal. ELISA results showed a concentration of 0.69 ng/mL ± 0.05 of FV in the concentrated secretome of WT HepG2, 0.57 ng/mL ± 0.1 of FV in the concentrated secretome of KO HepG2 and 0.55 ng/mL ± 0.1 of FV in the concentrated secretome of TT HepG2. No signal was detected on ELISA for the negative controls, the oval cell line or the SW480 cell line (Figure 5).

### 2.4. Functional FV Synthesized by the Three Cell Lines

The functional FV synthesized by the WT HepG2, KO HepG2 and TT HepG2 cell lines was tested by means of a viscosity-based coagulation assay, which revealed that the concentrated secretome from the WT HepG2 cell line contained around 108% ± 10.7 functional FV (sample coagulation time was ±21.3 s), while the concentrated secretome from the KO HepG2 cell line contained 19% ± 6.5 of functional FV (sample coagulation time was ±31.2 s). The concentrated secretome of the TT HepG2 cell line contained 31.5% ± 9 functional FV (sample coagulation time: ±27.7 s), which was significantly different from the percentages obtained with the KO HepG2 cell line (*p*-value = 0.00125). No significant differences were found between the KO HepG2 cell line and the negative controls (Figure 6). The increment in functional FV levels (ΔFV) between the TT HepG2 and KO HepG2 cell lines was 40.2% ± 3.9, which was in line with the 41.4% DNA correction achieved.

### 2.5. Analysis of Off-Target TT Guides

An analysis of genomic regions was made through Sanger sequencing, including the exons and introns of some genes (Table 1). No mutations or alterations in the sequence of these genomic regions were detected. This could be indicative of the fact that our designed guides were specific for the *F5* sequence, with no mutation or alteration being observed in the off-target genes analyzed.

## 3. Discussion

This study proposes a new advanced gene editing-based therapy for the correction of a new mutation in the *F5* gene (c.3279G>A, p.Trp1093*), previously described by our research group [24]. A specific CRISPR/Cas9 technology was developed to treat coagulopathies arising from this mutation, which is protected by a Spanish patent (Ref. ES2785323B2) with international coverage [25,26].

The study was devoted to the development of a custom cell model based on the mutation present in the patient [24], who had a severe FV deficiency (<1%). A strategy to correct this mutation and restore full production of functional FV was designed. The patient presented with two nonsense mutations (3279G>A, p.Trp1093* and 2218C>T, p.Arg740*), which produced premature stop codons giving rise to FV deficiency. It was decided to choose mutation 3279G>A, p.Trp1093* for the therapeutic model proposed because mutation c.2218C>T, p.Arg740* was close to a thrombin binding domain, which made it difficult to perform small deletions without disrupting the production of the KO cell model. Moreover, in heterozygous conditions, the Trp1093* mutation reduces FV plasma levels to 21% [24].

Developing a model of the kind proposed in this study is extremely useful given the difficulties inherent in working with samples from a patient with a hemorrhagic condition [27,28,29]. This model allows for testing custom treatments (in progress in our laboratory) for such patients (Figure 7).

HepG2 was selected for this study as it is a human immortalized cell line characterized by its robustness, durability, fast proliferation and potential to generate colonies from a single cell. It also expresses the *F5* in a constitutive manner in a hepatic cell line, which is advantageous considering that FV is synthetized in the liver [30,31]. Primary WT hepatocyte cultures are too unstable and short-lasting to carry out gene editing and clonal culture isolation studies such as those conducted as part of this analysis [32,33,34].

Another option would have consisted in obtaining hepatocytes from induced pluripotent stem cells (iPSc) from the studied patient, but, as previously reported by the authors of this study, iPSc or mesenchymal stem cell-derived hepatocytes cannot produce FV as the process requires a tridimensional environment [35,36]. 

The purpose of this study was to obtain a simple and efficient cell model to test different in vitro strategies for correcting *F5* mutations by gene editing or gene therapy using viral vectors.

The cell model used was developed through gene editing by means of a pair of guides (gRNA) based on the CRISPR/Cas9 technique. These guides direct the activity of the Cas9 protein to very specific sites in the genome [37], performing a precise deletion of the target sequences and an NHEJ repair. The use of two CRISPR gRNAs increases the efficacy of the NHEJ repair-mediated gene editing procedure. This pathway permits a higher editing efficacy than homologous repair using only a single guide. NHEJ-PD is based on the deletion of small DNA fragments through double DNA breaks mediated by two CRISPR guides. This typically gives rise to the removal of such fragments and to a perfect splicing of the two extremes. Apparently, any area in the genome may be subjected to CRISPR editing provided that it contains a PAM site (specific to each kind of Cas endonuclease used) and that the design of an RNA guide is possible [21]. The existence of numerous kinds of Cas endonucleases with different PAM sequences allows the formation of DNA double breaks anywhere in the genome and, consequently, the edition of any gene in the genome, with varying degrees of efficacy. This strategy is obviously faced with certain limitations associated with the design of the CRISPR guides used to induce double chain breaks. These include [22] the breaking distance between both guides. The optimal distance is between 23–148 bp. In general, a precise deletion is more likely when the distance between the guides is 30–500 bp. Orientation of the PAM sequence and of the sequence to be deleted within the guide is also very important; in/in orientations have been the most efficient for producing accurate deletions. It must be noted, however, that the distance between guides and the orientation of the PAM sequences have an impact on editing efficacy but not the ability to edit sequences. 

KO gRNAs allowed us to produce a highly efficient KO FV cell model with a nonsense codon stop mutation similar to the one exhibited by the studied patient, in approximately the same area of exon 13 and resulting in the same phenotype. The heterogeneous culture was purified and enriched by harvesting, selecting and expanding a series of clones until homogeneous cell lines were obtained. Finally, an analysis was made in three KO clonal FV cell lines.

The same gene editing strategy, based on TT gRNAs, was used to correct the mutation in the three clones of the KO FV cell line. This resulted in the restoration of the frameshift of the *F5* gene with an efficiency of 41% and in the generation of a functional FV cell line. The design of the guides and the PAM orientation influences the cleavage/splicing efficacy achieved by the NHEJ-PD strategy [22]. The guides used to induce the KO model (KO guide 5 and KO guide 1) (Figure 1B) were designed with the PAM sites facing the sequence to be deleted (in/in). Meanwhile, in the guides used to induce the correction (TT guide 4 and TT guide 2) (Figure 1D), PAM sites were outside the sequence (out/out), which results in a lower editing efficacy. This data having been obtained in optimal in vitro conditions, it must be noted that subsequent analyses with in vitro viral vectors using cell models, in vivo animal models or even clinical trials, will exhibit a lower efficacy given the complexity involved. In spite of that, it is to be expected that the functional FV levels obtained will be high enough to transform a severely FV deficient phenotype (<1% coagulation factor) into a moderately deficient FV phenotype (1–10% FV) or even a mild/asymptomatic FV phenotype (>10% FV) [10,11].

This strategy, which involves the deletion of different regions of exon 13, is similar to the one developed for the B domain (exon 14) of factor VIII [38], reducing its size and then packaging the B-domain deleted molecule into multiple viral vectors without loss of function [39].

Characterization of the three different cell lines (WT, KO and TT) did not show differences in cellular morphology, cellular proliferation or membrane molecular markers. These results seem to indicate that gene editing using the CRISPR/Cas9 method does not alter genes associated with the previously mentioned characteristics. 

The analysis of the off-target genes did not show any modification in the genetic sequences derived from the unspecific editing of the CRISPR guides used, at least for the genes studied. The analysis of possible off-target sequences was conducted using the TT HepG2 cell line edited with the two CRISPR guides simultaneously using Sanger sequencing and database sequence comparison (NCBI, Ensembl and HGMD).

Due to the nature of this cell line, it is difficult to perform an NGS analysis to detect potential DNA or chromosome rearrangements. Indeed, the HepG2 cell line exhibits an aberrant karyotype with multiple DNA sequence rearrangements, duplications, deletions, polyploidies, etc., making it impossible to evaluate the potential CRISPR-induced chromosome or DNA rearrangements as these cells per se harbor multiple DNA alterations [40].

In the immunostaining and ELISA assays, no significant differences were found in the fluorescence or concentration of FV between the three cell lines studied (WT, KO and TT). This could indicate that the three cell lines were responsible for the production and secretion of FV, as detected by polyclonal antibody-based immuno-staining assays and ELISA. In the case of the KO cell line, the coagulometry assay found that the FV produced was not functional. Our hypothesis is that the KO cell line can synthesize and secrete a stable fragment of FV (heavy chain), as detected by polyclonal antibody-based assays, but not a complete protein (without a light chain). This is in line with previous reports arguing for the need to use monoclonal antibodies to target the region of interest, as polyclonal antibodies are not able to differentiate between a full and a truncated protein just on the basis of an ELISA assay [41]. This means that viscosity-based coagulometric assays should be the method of choice to quantify and evaluate the functionality of FV, and Sanger sequencing to locate the pathological mutations [24,42,43,44]. 

The FV obtained is, therefore, not functional because it lacks the light chain, which plays a very important role in the coagulation pathway as it enables FV to bind to platelet membrane phospholipids [5,6].

By the same token, although no significant differences were observed in FV concentrations between the TT cell line and the WT and KO cell lines, the functionality of FV in the TT cell line was found to be significantly higher than in the KO cell line. This shows that CRISPR/Cas9-based gene editing can effectively correct the pathological mutation and produce an FV protein with renewed functionality. There was no difference between the KO cell line and the negative controls in the fully functional FV production. This difference in the functionality of FV between the KO and the TT cell lines (ΔFV) was proportional to the editing efficiency obtained by using TT gRNAs.

Confirmation of the results obtained by Western blot was not possible using any of the usual protocols and antibodies (Life Technology, Thermofisher Scientific, Abcam, Abbexa). Nor was it possible to find any study or protocol that produced positive Western blot results using the HepG2 cell line secretome. These persistently negative results are in line with those found by other authors, who have detected a 330 kda band in the plasma of patients with a truncated FV protein, corresponding to the whole protein, but have been unable to detect bands derived from a potentially truncated protein [41].

There are currently no deficient FV animal models that allow the performance of preclinical studies that can be used to test the efficacy of our TT gRNAs in correcting the mutation of the *F5* gene and restoring or increasing FV plasma levels in vivo. A few research groups have tried to develop deficient FV animal models, but such models have not been viable as none of the animals ever reach adulthood [45,46,47].

The next steps of our ongoing research will be focused on developing an FV-deficient animal model to test in vivo CRISPR-based gene editing and gene therapy protocols using lentiviral vectors.

In conclusion, a KO FV cell model with a mutation similar to the one carried by the patient studied was developed through NHEJ-PD with a CRISPR-based method. Moreover, an effective, precisely customized treatment, based on CRISPR gene editing, was developed to correct the mutation. The treatment proposed in this study succeeded in producing, at least in vitro, a functional FV protein. No molecular defect or alteration was found in the edited cells. This strategy could constitute a promising alternative for the development of custom pathologic models. 

## 4. Materials and Methods

### 4.1. Clinical Characteristics of the Patient

This study analyzed a 14-year-old Caucasian female from Jaen, in the south of Spain, afflicted with severe FV deficiency (FV plasma level < 1%). She was a compound heterozygote presenting with two different mutations: one inherited from her father and the other from her mother. Both parents exhibited a mild FV deficiency. The ensuing cell model and CRISPR/Cas9-driven correction studies centered on a new mutation (c.3279G>A, p.Trp1093*), previously described by our group [24].

### 4.2. Mutational Analysis

Genetic maps were developed with the SnapGene 1.1.3. software (San Diego, CA, USA), and the genetic sequence and protein sequence (with their respective domains) were obtained from the NCBI (NM_000130.5), Ensembl (ENST00000367797.9) and HGMD databases.

### 4.3. Cell Lines

A HepG2 cell line [30,31] (isolated from a hepatocellular carcinoma), made available by Dr. Segovia’s laboratory at CIEMAT, Madrid, Spain, was cultured in Dulbecco’s Modified Eagle Medium (DMEM; Thermofisher Scientific, Walpham, MA, USA), 10% fetal bovine serum (FBS; Biowest, Nuaillé, France) and 1% penicillin/streptomycin (Thermofisher Scientific, Walpham, MA, USA). Cells were seeded at a density of 5 × 10^4^ cells/cm^2^. Incubation conditions were 37 °C, 5% CO_2_ and 95% relative humidity.

To obtain a homogeneous KO FV cell line, 200 cells were seeded in a 100 cm^2^ Petri dish following the electroporation phase. The colonies that formed two weeks later were transferred to 24-well dishes in order to expand the clones and obtain different clonal cell lines.

### 4.4. Design and Generation of CRISPR-Cas9 Tools

To obtain the KO FV in the HepG2 cell line (KO HepG2) as well as the so-called “treated HepG2” (TT HepG2) cells, where the KO mutation was eliminated or reversed, recourse was had to specific CRISPR guides (gRNA) in a ribonucleoprotein (RNP) format. The different crRNAs (CRISPR RNAs) were designed by applying the “Zhang lab’s” software (MIT) to the sequence between nucleotides 3100 and 3400, located at *F5*’s exon 13. Selection of these crRNAs was based on the location and distance between those two crRNAs in the sequence to be edited and on the score corresponding to crRNAs in terms of their guanine and cytosine contents, their self-assembling potential, etc.

gRNAs were formed by the combination of tracrRNA (trans-activating crRNA) and crRNA on the basis of the following protocol: 0.4 µL of 200 µM tracrRNA and 0.4 µL of 200 µM crRNA were incubated at 95 °C for 5 min. When the solution reached room temperature, 0.2 µL of 62 µM nuclease Cas9 (*S. pyogenes*, IDT) was added, and the mixture was incubated for at least 20 min. All reagents were obtained from Integrated DNA Technologies (IDT, Coralville, IA, USA).

The guides used for correcting the mutation (TT gRNAs) were patented as an effective in vitro treatment by the Spanish Patent and Trademark Office (ES 2785323 B2). Table 2 shows the gRNAs used in the cell lines (KO gRNAs; TT gRNAs).

### 4.5. Electroporation of the HepG2 Cell Line with RNPs

A total of 2 × 10^5^ cells were added to a solution comprising 2 µL of each RNP (gRNA + Cas9) and 20 µL of “SF cell line Nucleofector^®^ solution” (Lonza, Basel, Switzerland). The resulting mixture was electroporated with an Amaxa 4D-Nucleofector system (Lonza, Basel, Switzerland), using a 20 µL volume under the EH-100 program.

### 4.6. Sequencing Analysis

The Sanger sequencing procedure was used to obtain the sequence of the different clones with a view to selecting (and subsequently expanding) the clone associated with the most accurate sequence. This sequencing method was used to determine the efficacy of the correction of the KO mutation and to analyze the sequences of potentially “off-target” genes as a way of establishing the degree of specificity of the guides. The Sanger sequencing procedure was carried out by the Genomics and Proteomics Unit of the Madrid Complutense University. 

In order to sequence the HepG2 cell lines using the Sanger method, total genomic DNA was extracted from the cells using the “DNeasy^®^ Blood and Tissue kit” (Qiagen, Germany) in accordance with the manufacturer’s instructions. The end-point PCR test (Biotools, Spain) was carried out using primers capable of amplifying 403 bp of WT HepG2; 368 bp of KO HepG2 and 319 bp of TT HepG2 of the human *F5* (which contains the target region in the middle of the amplicon). For “off-target” genes, different primers with different amplicon sizes were used. Table 3 shows the primers used to amplify the *F5* and the “off-target” genes, as well as the size of the amplicon.

The results of the Sanger sequencing assay were analyzed using the online version of Synthego’s ICE software (Synthego Corporation, Palo Alto, Santa Clara, CA, USA).

### 4.7. Characterization of HepG2 Cell Lines

A Zeiss Axiovert 40 C inverted microscope equipped with camera (Zeiss, Germany) was used to study the morphology of the different HepG2 cell lines. Images were captured with a 10× objective.

The proliferation index of the different HepG2 cell lines was evaluated using an Alamar blue assay (Alamar Blue™ Cell viability Reagent; Thermofisher Scientific, Walpham, MA, USA) in accordance with the protocol established by the manufacturer. A total of 1 × 10^4^ cells were briefly seeded in a 12-well dish with 10% FBS DMEM. The Alamar Blue reagent was subsequently diluted at 1:10 with 10% FBS DMEM. The solution was incubated for two hours, and a measurement was made at 560 nm excitation and 590 nm emission. This procedure was repeated daily for 7 days. 

Molecular membrane markers were evaluated through flow cytometry using the antibodies shown in Table 4. The flow cytometer was a BD FACSCanto II system (BD biosciences. San José, CA, USA). A total of 2 × 10^5^ cells were used to test each molecular membrane marker; antibody concentrations were those recommended by the manufacturer. Between 2 µL and 10 µL of each antibody was added.

### 4.8. Factor V Analysis and Quantification

Intracellular FV immunostaining was applied to HepG2 cells using Abbexa’s Coagulation Factor V (FV) antibody (Abbexa, Cambridge, UK) and the DyLight™ 488 Donkey anti-rabbit IgG (minimal x-reactivity) Antibody (Biolegend, San Diego, CA, USA) in accordance with the manufacturers’ instructions. Staining with 4′,6-diamidino-2-phenylindole (DAPI) was also carried out in order to identify the nuclei. Images were captured with a LeicaSP2 confocal microscope (Leica Biosystems, Barcelona, Spain). Human adipose-derived stem cells (hASC) were used as control.

An FV ELISA assay was performed with 6.5× concentrated secretome samples. The concentration of the secretome was achieved by using concentrator columns (Vivaspin^®^ 20, membrane: 100.000 MWCO PES (Sartorius. Stonehouse, UK) and by centrifuging 11 mL of secretome at 3000× *g* for 25 min. A WT HepG2 cell line was used as positive control, while mouse hepatic oval cells and the SW480 cell line were used as negative controls. The assay selected was the Human Factor V AssayMax ELISA Kit (Assaypro, St.Charles, MO, USA), which contained a polyclonal antibody. All the manufacturer’s instructions were followed. The data is shown as ng/mL of FV ± standard deviation.

A coagulation test was conducted using a STart Max II coagulometer (Stago, Barcelona, Spain) to ensure that the function of FV and its synthesis by different HepG2 cell lines were correct. FV function was tested using a 6.5× concentrated secretome from various cell lines. 

The coagulation assay was conducted following the manufacturer’s instructions. A magnetic steel ball, 50 µL of FV-deficient human plasma (Stago, Barcelona, Spain) and 50 µL of concentrated secretome were placed on a tray. The solution was incubated for around one minute in the coagulometer’s “incubation zone”, after which 100 µL of Neoplastine Cl plus (Stago, Barcelona, Spain) was added. When the sample coagulated, the device calculated the clotting time and extrapolated the data to a previously prepared standard curve. The data is shown as a % of functional FV ± standard deviation. The oval and the SW480 cell line-derived concentrated secretomes were used as negative controls to evaluate coagulation.

### 4.9. Off-Target Analysis

The potentially unspecific editions of the guides were tested by means of an analysis of the “off-target” genes using PCR assays and Sanger sequencing protocols. Two “off-target” sequence selection criteria were adopted to reduce the number of genes to be analyzed, thereby making the study feasible. One of the criteria involved an analysis of sequences as potential “off-target” sequences located in any of the gene’s exons (regardless of any differences in the number of nucleotides between the CRISPR guides and the gene). The second criterion consisted of regarding sequences located in any of the gene’s introns as potentially “off-target” provided that the difference in the number of nucleotides between the CRISPR guides and the gene was equal or less than 4. The analysis of the potential off-target editions was performed on the TT HepG2 cell line edited using the two CRISPR guides simultaneously (TT guide 4 and TT guide 2).

### 4.10. Data Analysis

The data were analyzed using the Statgraphics centurion XVII software package (Statgraphics Technologies Inc, The Plains, VA, USA). Statistical analyses were carried out by means of a t-test, which considered differences statistically significant when the *p*-value was below 0.05 (*), 0.01 (**) and 0.001 (***). The data were expressed as mean ± standard deviation.

## Figures and Tables

**Figure 1 ijms-23-05802-f001:**
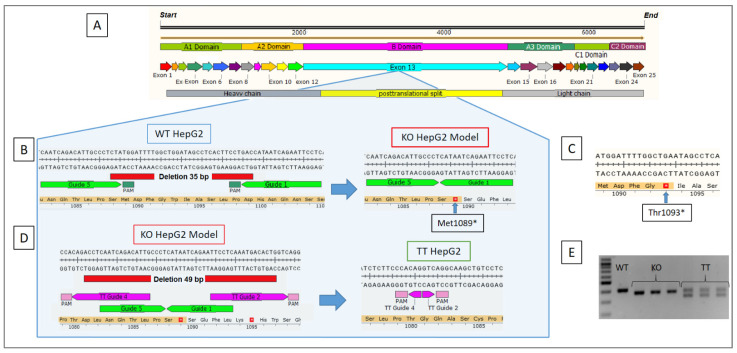
Gene editing strategy. (**A**) Structure of coagulation factor V. (**B**) KO HepG2 model development, simulating the similar mutation exhibited by the patient. (**C**) Patient’s Thr1093* pathological mutation. (**D**) Correction of the pathological mutation to obtain a treated HepG2 cell line (TT HepG2). (**E**) End-point PCR for the fragment of the *F5* gene on the wildtype HepG2 cell line, the three KO HepG2 clones (LO1, LO8 and LO9) and TT HepG2 (LO1, LO8 and LO9). A 100 bp FastGene molecular marker from Nippon genetics (Japan) was used.

**Figure 2 ijms-23-05802-f002:**
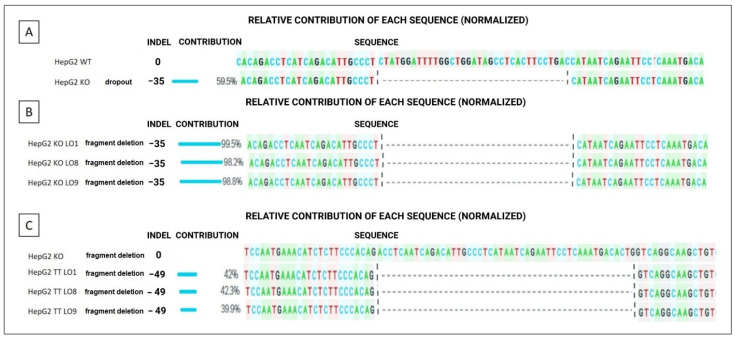
Sequencing of the different editing procedures performed in the *F5* of the HepG2 cell line. (**A**) Editing efficiency of the WT HepG2 line for obtaining the KO HepG2 line (by deleting 35 bp). (**B**) Three clone KO HepG2 cell lines (LO1, LO8 and LO9) were homozygous for the deletion of 35 bp, giving rise to the formation of a codon stop. (**C**) Editing efficiency of the KO clones for canceling the mutation and restoring the gene’s frameshift, giving rise to a corrected TT HepG2 cell line. Indel: insertion/deletion. Analyses were developed with the online version of Synthego’s ICE software.

**Figure 3 ijms-23-05802-f003:**
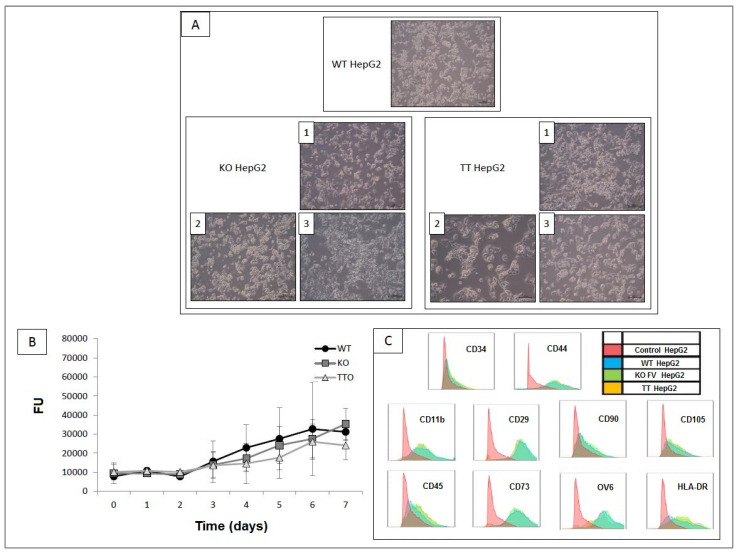
Characterization of different HepG2 lines. (**A**) Morphological changes of different clones (1, 2 and 3) (scale bar, 200 μm). Representative images from different experiments. (**B**) Proliferation curve presented as mean ± SD (*n* = 4) (*p* < 0.05). (**C**) Flow cytometry for membrane markers: Unstained cells (in red), WT HepG2 (in blue), KO HepG2 (in green) and TT HepG2 (in orange). Representative images from three different experiments.

**Figure 4 ijms-23-05802-f004:**
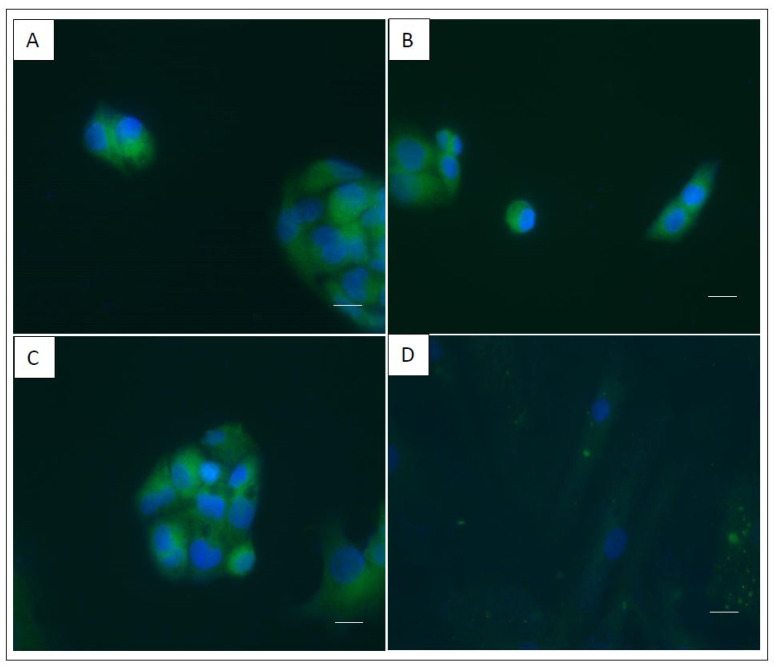
Coagulation factor V immuno-staining of cell lines. (**A**) WT HepG2. (**B**) KO HepG2. (**C**) TT HepG2. (**D**) hASCs as negative control (scale bar, 50 μm). Representative images from three different experiments.

**Figure 5 ijms-23-05802-f005:**
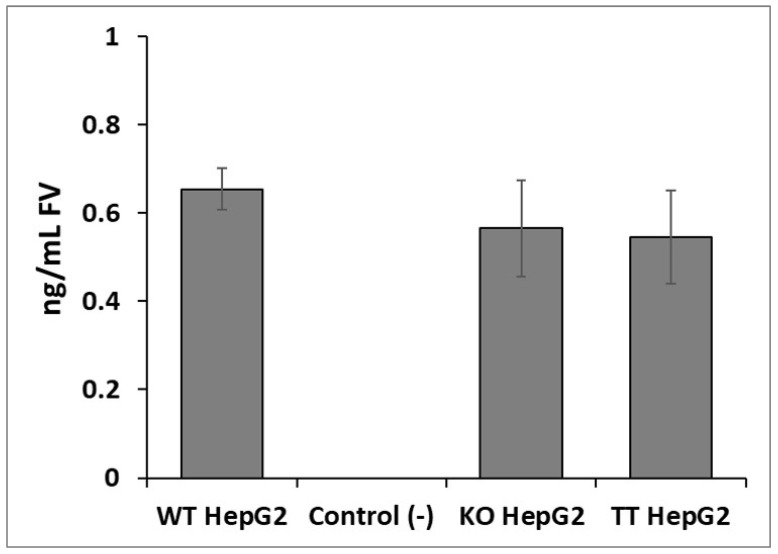
Application of the Human Coagulation Factor V ELISA assay to different HepG2 cell lines (WT HepG2, KO HepG2 and TT HepG2). Control (−) Oval cells and SW480 as negative controls. Figure presented as mean ± SD (*n* = 3).

**Figure 6 ijms-23-05802-f006:**
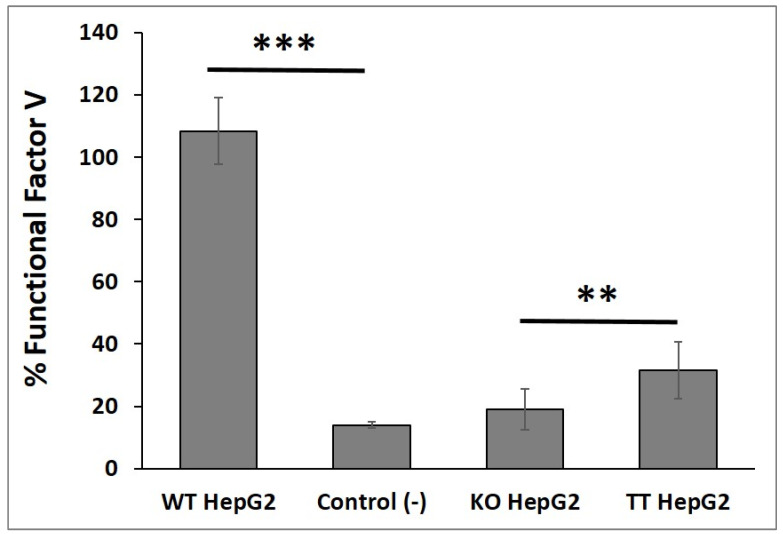
Functional factor V coagulometry assay. Functional factor V secreted by different cell lines (WT HepG2, KO HepG2 and TT HepG2). Control (-): Oval cells and SW480 as negative controls. Figure presented as mean ± SD (*n* = 4) (*** *p* < 0.001; ** *p* < 0.01).

**Figure 7 ijms-23-05802-f007:**
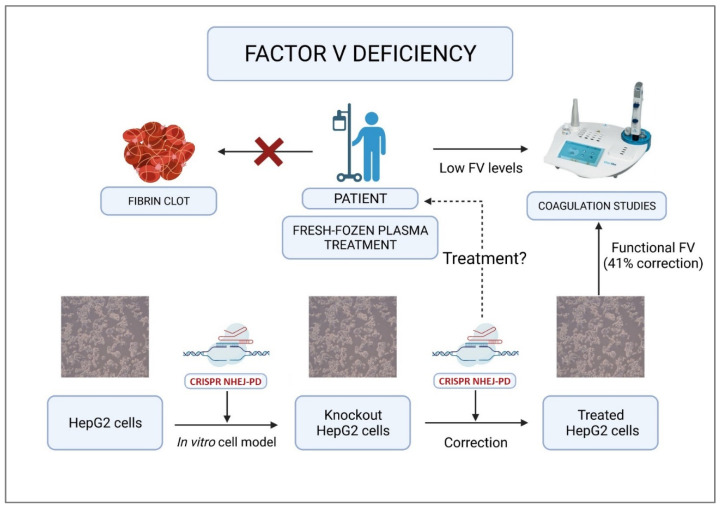
Summary of strategies and perspectives of the treatment of factor V deficiency. Using CRISPR/Cas9-based gene editing, the FV-producing HepG2 cell line was modified, giving rise to an in vitro knockout cell model unable to produce functional FV. The same tool was used to treat the modified cells and thereby correct the resulting mutation. This led to the recovery of 41% of FV’s functionality. This technology could constitute a gene therapy that could potentially cure FV deficiency.

**Table 1 ijms-23-05802-t001:** Off-target gene analysis.

*TT gRNA 2 Off-Targets*			
	*Gene*	*Bp*	*Mismatches **
EXON	SFTA3	409	5
	TFCP	396	5
	KIF9	387	5
	ABLIM3	496	4
	GAS7	464	5
INTRON	iGPC4	474	2
	iPRKCH	382	3
	iWDR4	492	3
	iWDFY4	385	3
	iZNRF3	403	3
	iEPC1	500	13
	iNRG3	423	4
** *TT gRNA 4 Off-Targets* **			
	** *Gene* **	** *Bp* **	** *Mismatches ** **
EXON	ARID1A	384	4
	RSPO2	350	5
	TREML1	394	5
	PPIL2	403	5
	GPR114	456	5
	LYPD6	405	4
	SCL25A3	472	5
	SKIL	382	4
	TENM4	440	4
INTRON	iRAD51B	405	2
	iMYO18	439	4
	iTEM8	388	4
	iEEFSE	411	4
	iFOXP1	402	4

* Different nucleotides between gRNA CRISPR and the gene sequences. ARID1A: AT-rich interaction domain 1A; RSPO2: R-spondin 2; TREML1: triggering receptor type 1 expressed on myeloid cells; PPIL2: peptidylprolyl isomerase type 2; GPR114: adhesion G protein-coupled receptor G5; LYPD6: Ly6/PLAUR domain; SCL25A3: mitochondrial phosphate carrier protein; SKIL: SKI-like proto-oncogene; TENM4: teneurin transmembrane protein 4; iRAD51B: intron of RAD51 paralog B (recombinase); iMYO18: intron of Myosin 18; iTEM8: intron of anthrax toxin receptor 1; iEEFSE: intron of Eukaryotic elongation factor, selenocysteine-tRNA-specific; iFOXP1: intron of Forkhead box P1; SFTA3: surfactant-associated 3; TFCP: transcription factor CP; KIF9: kinesin family member 9; ABLIM3: action-binding LIM protein family member 3; GAS7: growth arrest-specific 7; iGPC4: intron of Glypican 4; iPRKCH: intron of protein kinase C eta; iWDR4: intron of WD repeat domain 4; iWDFY4: intron of WD repeat and FYVE domain-containing protein 4; iZNRF3: intron of zinc and ring finger 3; iEPC1: intron of enhancer of polycomb homolog 1; iNRG3: intron of neuregulin 3.

**Table 2 ijms-23-05802-t002:** gRNAs used in the HepG2 cell lines (KO gRNAs, TT gRNAs).

*Guide*	*Sequence*	*PAM*
KO Guide 5	CAATCAGACATTGCCCTCTA	TGG
KO Guide 1	GAGGAATTCTGATTATGGTC	AGG
TT Guide 4	CAATGTCTGATTGAGGTCTG	TGG
TT Guide 2	TTCCTCAAATGACACTGGTC	AGG

gRNA: RNA guide; KO: *Knockout model* HepG2; TT: treated HepG2.

**Table 3 ijms-23-05802-t003:** Primers used to amplify *F5* and off-target genes.

*Primers*				*Bp Product*		
*Gene*			*Sequence 5′-3′*	*WT*	*KO*	*TT*
Factor V	Forward	GAGAAGCACACACACCATGC		403	368	319
	Reverse	CTGTGGGGAAGGACTTGTGA				
** *TT gRNA 2 off-targets* **						
	** *Gene* **		** *Primer 5′-3′* **			** *bp* **
EXON	SFTA3	Forward	CCAACAAGGGTACACACTGC			409
		Reverse	TCTCAGAATTCCTCCCTGCG			
	TFCP	Forward	CAGCCTTCCTATGAGACAACCA			396
		Reverse	GAACCAAAGGCAGCACACAG			
	KIF9	Forward	GCTCCTGTCTTGGGTAGGTGG			387
		Reverse	CCAAAGCTCCCTAGCCCTCTT			
	ABLIM3	Forward	CAGGCTATGTAGTCCCTGAGC			496
		Reverse	ATCACCTGCAGAGGACCCAA			
	GAS7	Forward	ATGCCTGGAGAGGTCAGGAT			464
		Reverse	CAGCCCTGCCATACACAAGA			
INTRON	iGPC4	Forward	GACCTTGACCCCAGACAACT			474
		Reverse	TTTGAGGGGGAGGAAGTGTTC			
	iPRKCH	Forward	CACTGATTTCACCCTTAGCAAACC			382
		Reverse	TAGGGCGAAGCTTTACCTGG			
	iWDR4	Forward	AGGTTGGCCTTGACTTTGGG			492
		Reverse	CCAGTGTCCGAAAGGCAAAATG			
	iWDFY4	Forward	ATAGAGCATGTGGAGCCCTTG			385
		Reverse	GGGAAGGGAGAAGGGCATTG			
	iZNRF3	Forward	GGATCTCTTACCAGCCCCCT			403
		Reverse	GATCCTCCACTTGCTGATGG			
	iEPC1	Forward	TCTTGGAAAGTGTACATCTGAGG			500
		Reverse	CAGGCTTGAAGGTCTGTTTACC			
	iNRG3	Forward	TGATGTTGTAGCCTAGGGAAGT			423
		Reverse	AGCAAACCTCTGCATCCATCT			
** *TT gRNA 4 off-targets* **						
	** *Gene* **		** *Primer 5′-3′* **			** *bp* **
EXON	ARID1A	Forward	TGGGAAAGGAGCAACTCTGC			384
		Reverse	TGTCCTTGCCTGAAAAGGCT			
	RSPO2	Forward	AGCAATGGGCAGAGCCATTC			350
		Reverse	ATGTGAACCACACCTAACCTGA			
	TREML1	Forward	GAGCCAGTGCCATTTCCTGA			394
		Reverse	CCCCTTCGATTTAGTTTGGAGC			
	PPIL2	Forward	CTGTTGTGCCACAGATTGCG			403
		Reverse	ATACAGCCCTCCCGGTGTAG			
	GPR114	Forward	CAAAGATTGGGAGGTTCCGC			456
		Reverse	AATGGGTCACTCAGTAGGCG			
	LYPD6	Forward	AGGTCTGCACTTCTTGTTGTG			405
		Reverse	TGGTATGGAACTTGGGCTGT			
	SCL25A3	Forward	CGCAGGCATTTTTCCCTTGC			472
		Reverse	GTCTGACATTCGCTCTTTACATGG			
	SKIL	Forward	AAGTAGTCAGAGCTCGCTGG			382
		Reverse	TGTAATCAGCCCACAGGATGG			
	TENM4	Forward	GGCCCTTTTGACACTCACCA			440
		Reverse	CACCTCCTGGCCATTCTCAC			
INTRON	iRAD51B	Forward	TCTGCCACTCCTGACTTCCT			405
		Reverse	TCTTCCTAGTCACCTCGCCT			
	iMYO18	Forward	TAGGGCTCATTGCAAGGAAGAT			439
		Reverse	CCCCAGCCTCCAATCTGAC			
	iTEM8	Forward	GCCAAGCGTGTAGCAGACT			388
		Reverse	TGGATGTCTGCCAGCAAAGG			
	iEEFSE	Forward	GAAGACCTGCCTCTTCCTGA			411
		Reverse	CCCTCCCCTTACCTTTTCCAC			
	iFOXP1	Forward	TGATAGAACCTCGCTTGGGG			402
		Reverse	TAGTCTCTGGACTGGTGTGGG			

gRNA: RNA guide; KO: *Knockout model* HepG2; TT: treated HepG2; WT: wildtype HepG2. gRNA: RNA guide; SFTA3: surfactant-associated 3; TFCP: transcription factor CP; KIF9: kinesin family member 9; ABLIM3: action-binding LIM protein family member 3; GAS7: growth arrest-specific 7; iGPC4: intron of Glypican 4; iPRKCH: intron of protein kinase C eta; iWDR4: intron of WD repeat domain 4; iWDFY4: intron of WD repeat and FYVE domain-containing protein 4; iZNRF3: intron of zinc and ring finger 3; iEPC1: intron of enhancer of polycomb homolog 1; iNRG3: intron of neuregulin 3; ARID1A: AT-rich interaction domain 1A; RSPO2: R-spondin 2; TREML1: triggering receptor expressed on myeloid cells like 1; PPIL2: peptidylprolyl isomerase-like 2; GPR114: adhesion G protein-coupled receptor G5; LYPD6: Ly6/PLAUR domain-containing 6; SCL25A3: mitochondrial phosphate carrier protein; SKIL: SKI-like proto-oncogene; TENM4: teneurin transmembrane protein 4; iRAD51B: intron of RAD51 paralog B (recombinase); iMYO18: intron of myosin 18; iTEM8: intron of anthrax toxin receptor 1; iEEFSE: intron of Eukaryotic elongation factor, selenocysteine-tRNA specific; iFOXP1: intron of Forkhead box P1.

**Table 4 ijms-23-05802-t004:** Antibodies used for flow cytometry.

*Antibody*	*Manufacturer*	*Fluorochrome*	*Concentration*	*Ref.*
CD90	Biolegend	PE	50 µg/mL	328,110
CD29	Biolegend	PerCP-Cy5.5	400 µg/mL	303,024
CD73	Biolegend	PE	400 µg/mL	344,004
CD45	Biolegend	FITC	200 µg/mL	304,006
CD34	Biolegend	APC	50 µg/mL	343,608
HLADR	Biolegend	APC	200 µg/mL	307,658
OV6	Santa Cruz Bio.	PE	200 µg/mL	SC-101863
CD11b	Biolegend	PerCP-Cy5.5	200 µg/mL	301,328
CD105	Biolegend	APC	100 µg/mL	323,208
CD44	Biolegend	PE/Cy7	100 µg/mL	338,816

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
