# Peer review of "Development and Characterization of a Factor V-Deficient CRISPR Cell Model for the Correction of Mutations"

_ijms, 2022, doi:10.3390/ijms23105802_

Round 1
Reviewer 1 Report
The article entitled “Development and characterization of a factor V-deficient CRISP cell model for the correction of mutations” describes a genetic therapy to cure a hereditary coagulopathy drug-orphan such as the Factor V deficiency. The project study and the aim are well described. This reserach study has an important translational clinical significance. The authors obtained normal concentratio of factor V using three cellular lines (WT, KO, and TT) and evaluated the activity of fcator V by coagulometer but the coagulation parameters (prothrombin activity, prothrombin time, INR) are not reported in the text. I think that this information is important and deserves to be written. Therefore, I think that this manuscript is suitable for publication if the authors mention this missing information.
Author Response
Dear Editor,
I am enclosing a point-by-point response to the referees’ comments. Changes to the text of the revised manuscript have been marked in red using the “track changes” function.
Point-by-point answers in this document are also marked in red.
Reviewer #1 comments
The article entitled “Development and characterization of a factor V-deficient CRISP cell model for the correction of mutations” describes a genetic therapy to cure a hereditary coagulopathy drug-orphan such as the Factor V deficiency. The project study and the aim are well described. This research study has an important translational clinical significance. The authors obtained normal concentration of factor V using three cellular lines (WT, KO, and TT) and evaluated the activity of factor V by coagulometer, but the coagulation parameters (prothrombin activity, prothrombin time and INR) are not reported in the text. I think that this information is important and deserves to be written. Therefore, I think that this manuscript is suitable for publication if the authors mention this missing information.
We thank the reviewer for their interesting remarks. Prothrombin time (PT) and INR as measured in routine blood work usually provide useful information. However, this is not the case in this instance as the measurements were made in a cell culture medium. Both PT and INR help estimate the overall performance of the extrinsic coagulation pathway, which requires the involvement of platelet membrane and vascular endothelial phospholipids, tissue factor and factor VII, as well as factors V, X, prothrombin, fibrinogen, calcium and vitamin K.
Vitamin K, supplied by the diet, is an essential cofactor to determine PT and INR as it is instrumental for the activation (by gamma-carboxylation of glutamic acid residues) of coagulation factors II, VII, IX and X, as well as anticoagulation proteins C, S and Z [Brenner, B.; Kuperman, A.; Watzka, M.; Oldenburg, J. Vitamin K–Dependent Coagulation Factors Deficiency. Semin. Thromb. Hemost. 2009, 35, 439-46. https://doi.org/10.1055/s-0029-1225766 // Dahlbäck, B. Vitamin K–Dependent Protein S: Beyond the Protein C Pathway. Semin. Thromb. Hemost. 2018, 44, 176-84. https://doi.org/10.1055/s-0037-1604092].
These physiological conditions are only present in blood vessels and are very difficult to reproduce in a cell culture. An evaluation of PT and INR would yield values far above the reference values of the three HepG2 cell lines evaluated (WT, KO and TT). This has been shown to be the case in patients with vitamin K deficiency but with no dysfunctions in any of their coagulation factors [Mullier, F.; Paridaens, M.S.; Evrard. J.; Baudar, J.; Guldenpfennig, M.; Devroye, C.; Miller, L.; Chatelain, B.; Lessire, S.; Jacqmin, H. Evaluation of a new thromboplastin reagent STA‐NeoPTimal on a STA R Max analyzer for the measurement of prothrombin time, international normalized ratio and extrinsic factor levels. Int. J. Lab. Hematol. 2020, 42, 650-60. https://doi.org/10.1111/ijlh.13236].
It is possible, however, to determine the functional levels of factor V and other coagulation factors as those are secretion proteins present in the culture medium. The concentration of such proteins can be measured independently using in vitro methods, or through a coagulometer ensuring that the conditions required for the assay are met.
Reviewer #2 comments:
The manuscript of Serrano and colleagues reports the development and characterization of factor V-deficient cellular model through the exploitation of a CRISPR/Cas9-driven homologous end-joining precision deletion (NHEJ-PD) approach which, in turn, can be further tailored to restore proper factor V expression. Authors exploited NHEJ-PD to introduce, in the HepG2 cells, deletion of 35bp (c.3263-3298del) thus modifying the frameshift of the gene sequence and giving rise to a premature stop codon at p.Met1089*. This change is in the proximity of the c.3279G>A, p.Trp1093*, one of the mutations exhibited by the studied patient. Then, NHEJ-PD was further exploited to produce a deletion of 49 bp (c.3242-3291del), thus removing the premature stop codon at p.Met1089* and re-establishing the correct FV reading frame. The rescue, albeit not clearly evident at antigen level, was observed as an increment in functional FV levels by the viscosity-based coagulation assay. The correction at functional FV levels was in line with the correction observed at DNA level.
Overall, the study is well conducted, but some major and minor points still need to be addressed:
- It is not clear why the authors do not create a cellular model harboring the mutation under investigation, the c.3279G>A, p.Trp1093* but, instead, with the p.Met1089* resulting from the deletion of c.3263-3298del. By using the CRISPR/Cas9 (with ssDNA as donor template) approach followed by the clonal analysis, authors had the possibility to create the proper HepG2 cellular model.
The main goal of this research was to generate a KO cell model for factor V that did not produce functional factor V as this would allow evaluation of the different advanced therapy protocols being designed in our lab. The gene editing protocol used was aimed at achieving higher editing precision. The ultimate purpose was to mimic the subject’s phenotypic features using a mutation of a similar kind, located in a similar position and which produced the same phenotype.
Although less precise than HDR, the NHEJ-PD strategy was used because it allowed fast and highly accurate generation of the models, robust editing control and the high efficacy and efficiency required when using this kind of repair pathway. It also permitted a close relationship between efficacy and cost, in terms of obtaining a pathological model [Ryu, S.M.; Hur, J.W.; Kim, K. Evolution of CRISPR towards accurate and efficient mammal genome engineering. BMB Rep. 2019, 52. 475-81. https://doi.org/10.5483/BMBRep.2019.52.8.149 // Guo, T.; Feng, Y.L.; Xiao, J.J.; Liu, Q.; Sun, X.N.; Xiang, J.F.; Kong, N.; Liu, S.C.; Chen, G.Q.; Wang, Y.; Dong, M.M.; Cai, Z.; Lin, H.; Cai, X.J.; Xie, A.Y. Harnessing accurate non-homologous end joining for efficient precise deletion in CRISPR/Cas9-mediated genome editing. Genome Biol. 2018, 19, 170. https://doi.org/10.1186/s13059-018-1518-x].
The consequence of this was an alteration in the production of non-functional factor V and the generation of a KO cell line amenable to be corrected by genetically editing the mutation produced.
- The observed efficiency of NHEJ-PD in HepG2 has been obtained upon electroporation of Cas9 RNPs. This approach (electroporation) is not translatable in vivo, so overall lower efficiency is expected. At this stage, where no viral vectors have been exploited to deliver Cas9 actors, the therapeutic potential of this approach should be smoothened and further discussed in the discussion section. Therefore, albeit the proof-of-principle has been provided, the overall soundness is lower than that mentioned.
We thank the reviewer for this interesting comment related to the potential clinical translation of our findings. The main goal of this study was to develop a mutated factor V cell model to be used for in vitro evaluations of future advanced therapy applications. This has been clarified and spelled out in the text (lines 244-250). In this regard, a phrase previously included in line 313 has been removed to avoid misunderstandings.
- Why the authors observed a NHEJ-PD efficiency of 60% during the generation of the HepG2 KO cell models, and only 40% during the rescue to generate the TT cellular models? This aspect has not been discussed properly.
This important comment by the reviewer deserves a more detailed explanation of the mechanism of action, the design, and the efficacy of the gene editing tool presented in the paper. As mentioned in the manuscript, and in line with the findings of the study in reference #22, the design of the guides and the orientation of the PAM sequences impact the cleaving/splicing efficacy of the NHEJ-PD strategy. As regards the guides used to induce the KO model (KO guide 5 and KO guide 1) (Figure 1B), they were designed with their PAM sites facing the inner part of the sequence to be deleted (in/in), while in the guides used to induce the correction (TT guide 4 and TT guide 2) (Figure 1D) the PAM sites face the outer part of the sequence (out/out), which means that – in the latter case – editing efficacy is lower than that of the mutation-producing gene editing process. This has been clarified in the text (lines 215-230; 239-244).
- Why authors did not create a homozygote TT FV cell line clone?
We did not contemplate the idea of obtaining a homozygous cell line with the reversal of the mutation as testing the real efficacy of the reversal of the mutation through CRISPR gene editing was only a secondary goal of this project. Obtaining such a cell line would not make such a big difference as the subsequent therapeutic approaches we intend to test in the future will be based on the homozygous KO line. The TT cell line was included just to evaluate the efficacy of the TT guides developed to correct the mutation.
- In the introduction, it is not clear which mutation will be targeted.
The reviewer is right. The text has been amended to clarify this point (lines 71-73).
- It is not clear, in the actual form, how the coagulation assay was conducted. It is stated that the assay was performed on concentrated (6.5X) cell media, but no information on FV antigen levels in these samples are provided. Does the reported time take into account the FV antigen levels in the concentrated media? Moreover, by looking at the SD in figure 6, it seems that KO HepG2 are not statistically different from TT HepG2.
Similarly, to the coagulometric analysis, the ELISA assays were conducted with 6.5X concentrated secretome samples. This has been specified in the text (lines 402-405). To avoid potential misunderstandings, the paragraph in lines 413-415 has been removed.
To address the reviewer’s remark about the statistical non-significance of the difference between the KO HepG2 and TT HepG2 lines, we are attaching a table that includes the data obtained from the study as well as a scatter plot showing that, although both cell lines display the same standard deviation, there is a significant difference between them, with a p-value of 0.00125 (according to the one-way t-test).
|
KO HepG2 |
TT HepG2 |
||||
|
LO1 |
LO8 |
LO9 |
LO1 |
LO8 |
LO9 |
|
18* |
7 |
17 |
35 |
13 |
31 |
|
21 |
16 |
24 |
42 |
24 |
38 |
|
27 |
11 |
22 |
36 |
24 |
30 |
|
27 |
42 |
||||
*Raw data from absolute measurements (% functional FV) in the KO and TT HepG2 cell lines.
- Authors should better discuss the limitation of NHEJ-PD approach. All genes can be virtually targeted by NHEJ-PD? In which circumstances this approach is and is not feasible?
We agree with the reviewer that clarification of these aspects may add to the quality of the manuscript. Accordingly, a new paragraph has been added to the text (lines 215-230; 239-244) to clarify and elaborate on the matter. Reference #21 has been updated (line 533).
- The actual off-target analysis is mainly focused on the possible effects of each single gRNA on other targets. However, when two gRNAs are exploited, off-target analysis should take into account genes targeted by both gRNA as well as DNA and chromosome rearrangements, not detectable by the actual analysis. Only a NGS analysis could detect all off-target effects of this approach. This limitation should be properly discussed in the text.
A small paragraph has been included (lines 261-263; 433-434) to clarify the methodology used for the off-target analysis.
This study did not include an analysis of the potential off-target effects on cells edited with one single CRISPR guide. These analyses have been performed, however, on the TT HepG2 cell line, which was edited with the two CRISPR guides at the same time. To discuss the value of analyzing potential off-target sequences, as well as the limitations of NGS analysis, a new paragraph has been added to the Discussion section (lines 264-268) and a new reference was included (new #40) (lines 268; 595) relative to the use of NGS analysis for evaluating potential off-target effects or DNA alterations for this specific cell line. Consequently, the order of references #40 to #47 (lines 268; 278; 280; 298; 303; 595-624) has been changed.

Reviewer 2 Report
The manuscript of Serrano and colleagues reports the development and characterization of factor V-deficient cellular model through the exploitation of a CRISPR/Cas9-driven homologous end-joining precision deletion (NHEJ-PD) approach which, in turn, can be further tailored to restore proper factor V expression. Authors exploited NHEJ-PD to introduce, in the HepG2 cells, deletion of 35bp (c.3263-3298del) thus modifying the frameshift of the gene sequence and giving rise to a premature stop codon at p.Met1089*. This change is in the proximity of the c.3279G>A, p.Trp1093*, one of the mutations exhibited by the studied patient. Then, NHEJ-PD was further exploited to produce a deletion of 49 bp (c.3242-3291del), thus removing the premature stop codon at p.Met1089* and re-establishing the correct FV reading frame. The rescue, albeit not clearly evident at antigen level, was observed as an increment in functional FV levels by the viscosity-based coagulation assay. The correction at functional FV levels was in line with the correction observed at DNA level.
Overall, the study is well conducted, but some major and minor points still need to be addressed:
1-it is not clear why the authors do not create a cellular model harboring the mutation under investigation, the c.3279G>A, p.Trp1093* but, instead, with the p.Met1089* resulting from the deletion of c.3263-3298del. By using the CRISP/Cas9 (with ssAON as donor template) approach followed by the clonal analysis, authors had the possibility to create the proper HepG2 cellular model.
2-the observed efficiency of NHEJ-PD in HepG2 has been obtained upon electroporation of Cas9 RNPs. This approach (electroporation) is not translatable in vivo, so overall lower efficiency is expected. At this stage, where no viral vectors have been exploited to deliver Cas9 actors, the therapeutic potential of this approach should be smoothened and further discussed in the discussion section. Therefore, albeit the proof-of-principle has been provided, the overall soundness is lower than that mentioned.
3-Why the authors observed a NHEJ-PD efficiency of 60% during the generation of the HepG2 KO cell models, and only 40% during the rescue to generate the TT cellular models? This aspect has not been discussed properly.
4-why authors did not create a homogenous TT FV cell line clone?
5-in the introduction, it is not clear which mutation will be targeted.
6-it is not clear, in the actual form, how the coagulation assay was conducted. It is stated that the assay was performed on concentrated (6.5x) cell media, but no information on FV antigen levels in these samples are provided. Does the reported time take into account the FV antigen levels in the concentrated media? Moreover, by looking at the SD in figure 6, it seems that KO HepG2 are not statistically different from TT HepG2.
7-authors should better discuss the limitation of NHEJ-PD approach. All genes can be virtually targeted by NHEJ-PD? In which circumstances this approach is and is not feasible?
8-the actual off-target analysis is mainly focused on the possible effects of each single gRNA on other targets. However, when two gRNAs are exploited, off-target analysis should take into account genes targeted by both gRNA as well as DNA and chromosome rearrangements, not detectable by the actual analysis. Only a NGS analysis could detect all off-target effects of this approach. This limitation should be properly discussed in the text.
Author Response

(The authors gave the same response as above.)

Round 2
Reviewer 2 Report
Authors fulfilled all my concerns.